# Geographic Area of Collection Determines the Chemical Composition and Antimicrobial Potential of Three Extracts of Chilean Propolis

**DOI:** 10.3390/plants10081543

**Published:** 2021-07-28

**Authors:** Marysol Alvear, Estela Santos, Felipe Cabezas, Andrés Pérez-SanMartín, Mónica Lespinasse, Jorge Veloz

**Affiliations:** 1Department of Chemical Sciences and Natural Resources, Faculty of Engineering and Sciences, La Frontera University, Francisco Salazar Avenue 01145, Temuco 4780000, Chile; marysol.alvear@ufrontera.cl (M.A.); aperez09@ufromail.cl (A.P.-S.); 2Department of Ethology, Faculty of Sciences, La Republica University, Iguá 4225, Montevideo 11400, Uruguay; esantos@fcien.edu.uy; 3Department of Biological and Chemical Sciences, Faculty of Medicine and Science, San Sebastian University, Campus Los Leones, Lota 2465, Providencia 7500000, Chile; felipe.cabezas@uss.cl (F.C.); monica.lespinasse@uss.cl (M.L.)

**Keywords:** chilean propolis, polyphenols, flavonoids, botanical origin, natural antimicrobials

## Abstract

The biological properties of chilean propolis have been described and include antibacterial, antifungal and antibiofilm activities. Propolis has a strong antimicrobial potential. Clinical experiences with synthetic antibiotics indicated the need to discover new sources of bioactive compounds associated with ethnopharmacological knowledge or natural sources such as propolis. The microscopic analysis of pollen grains from plants allows us to determine the botanical origin of the propolis samples. In Angol, sample pollen grains were obtained from fodder plants (*Sorghum bicolor; Lotus sp.*) and trees, such as *Acacia *sp.*, Pinus radiata, Eucalyptus *sp.** and *Salix babylonica*. Propolis from the Maule region contains pollen grains from endemic plants such as *Quillaja saponaria*. Finally, the sample obtained from Melipilla presented a wider variety of pollen extracted from vegetable species.Colorimetric assays performed to quantify the total polyphenols present in Chilean propolis samples established that PCP2 (Angol sample) showed high amounts of phenolics compounds, with significant statistical differences in comparison with the other samples. The main compounds identified were pinocembrin, quercetin and caffeic acid phenethyl ester (CAPE). The Angol sample showed a high content of polyphenols.Studies that determine the influence of geographical and floral variables on the chemical composition of propolis are a valuable source of information for the study of its biological properties.

## 1. Introduction

Propolis is a bee product, a resinous substance that contains secondary plant metabolites including volatiles metabolites; however, it is obtained from different vegetable species and is not the same anywhere in the world [1].Insects, such as honeybees, produce propolis using different parts of the plants and by mandibular secretions, and use it to embalm dead insects and to prevent the proliferation of microorganisms in the colony [2].

In addition to the chemical variability in its composition, largely depending on plant sources, this is also determined by the different geographic origin of the propolis samples. Therefore, the biological activities of propolis will depend on the polyphenols present in the samples. Propolis typically consists of 50% plant resins, 30% waxes, 10% essential and aromatic oils, 5% pollen and 5% other organic substances [3]. A broad spectrum of biological properties has been described for the extracts of propolis, such as antidiabetic, anti-atherogenic, antibacterial and antifungal properties, which are mainly related to the polyphenols content of the propolis samples [4,5,6].

Clinical experiences with synthetic antibiotics indicated the need to discover new sources of bioactive compounds associated with ethnopharmacological knowledge.In this context, resistance to antibacterial compounds is an important problem worldwide and the correct isolation and identification of bioactive compounds from natural products is required. However, the selection of the species must be based on popular medical indication, isolation and determination of active substances and on the choice and implementation of pharmacological assays [7].

The search for new antibacterial agents has been prompted by the increase in bacterial resistance, caused by drug interactions and adverse effects, as well as the indiscriminate and excessive use of traditional antibiotic drugs.To this end, natural products have been an important source for the identification of new active molecules. For this reason, in the same way as other natural products such as broccoli, cocoa, and laurel, propolis has been studied in order to explore its antigenotoxic potential.This property was first investigated in 2005 [8,9]. The purpose of this study is to determine the influence of the geographic area of collection on the polyphenol content and antimicrobial activity of the chilean propolis extracts.

## 2. Results

### 2.1. Botanical Origin of Chilean Samples

The microscopic analysis of pollen grains and other fragments of plant debris left by bees is a technique that allows to determine the botanical origin of propolis samples. In the Angol sample (PCP1), pollen grains were obtained from fodder plants(*Sorghum bicolor; Lotus *sp.**), and from other plants such as *Cichorium intybus;*
*Raphanus raphanistum*. In addition, propolis from Angol showed pollen grains from shrubs and trees, such as *Acacia *sp.*, Pinus radiata, Eucalyptus *sp.** and *Salix babylonica*.In the same way, in the PCP2 sample from the Maule region we identified pollen grains from endemic plants such as *Quillaja saponaria*, besides grains from some invasive plants, for example *Genista monspessulana* (French broom).Finally, the PCP3 sample obtained from Melipilla showed a greater pollen variety extracted from vegetable species such as *Salvia officinalis*, *Scutia buxifolia* (Coronilla);other pollens were obtained from tree species such as *Casuarina cunninghamiana*; *Ligustrum lucidum* and *Eucalyptus sp*. The PCP3 sample exhibited a greater variety of vegetables specie. Table 1, shows the predominant species identified in the PCP1, PCP2 and PCP3 samples of propolis.

### 2.2. Total Polyphenols Content in Chilean Propolis and Compounds Quantification

In the propolis extract (PE),the total polyphenols content in equivalence of pinocembrin–galangin mixture was quantified by the Folin–Ciocalteu reaction, obtaining the following results: PCP1 51,530 ± 1 mgg^−1^; PCP2 45,269 ± 4 mgg^−1^; and PCP3 28,696 ± 2 mgg^−1^.

The entire total flavonoids content (flavones and flavonols), quantified by AlCl_3_ methodology was: PCP1 19,331 ± 2 mgg^−1^; PCP2 16,440 ± 4 mgg^−1^; and PCP3 12,297 ± 5 mgg^−1^. The Angol sample exhibits significant statistical differences in comparison with the other samples, as shown in Figure 1.

On the other hand, the total content of flavanones and dihydroflavonols, determined by reaction with 2,4-dinitrophenylhydrazine, was in PCP1: 31,397 ± 3 mgg^−1^; in PCP2: 35,390 ± 2 mgg^−1^ and in PCP3: 22,255 ± 3 mgg^−1^. The Maule sample exhibits significant statistical differences in comparison with the other samples, as shown in Figure 2. The mean percentages of the total phenolic compounds means were similar to the results of previous studies [10,11,12].

The content of total tannins in PCP1 was 7807 ± 1 mgg^−1^; in PCP2 was 8311 ± 2 mgg^−1^ and in PCP3 was 5719 ± 4 mgg^−1^. Anthocyanins were quantified and in PCP1 the value obtained was 4145 ± 5 mgg^−1^; in PCP2 was 3867 ± 4 mgg^−^^1^ and in PCP3 was 2809 ± 7 mgg^−1^. Tannins and anthocyanins did not show statistical differences between the three samples, as shown in Figure 2.

The main flavonoids found in chilean propolis were identified using HPLC-DAD, as shown in Figure 3. According to the direct comparison with commercial standards and the time of retention (Tr), the pinocembrin was verified as the most abundant compound present. In the three samples we identified the presence of other polyphenols such as galangin, quercetin, apigenin and caffeic acid phenethyl ester (CAPE); these results were similar to previous studies [13,14,15].

HPLC quantification of the main individual compounds is shown in Table 2. Pinocembrin was quantified over 107.3 mg L^−1^ in all samples. Quercetin was abundant, with concentration values between 88.8 to 127.9 mg L^−1^. Apigenin, galangin and CAPE were present in low concentrations.PCP3 had the lowest concentrations of CAPE, pinocembrin and quercetin, with statistical differences in comparison with the Angol and Maule samples.

### 2.3. Balsam Content in the PCP1,PCP2 and PCP3 Samples of Chilean Propolis

In PE, the mean values of ethanol-soluble balsams were determined. Balsam contents were high: in PCP3 was 0.70–0.69 g (56.3%) in comparison with PCP2 (0.49 g, 52.2%) and in PCP1 (0.69 g, 53.3%), within statistical differences. 

### 2.4. Test for Determining the Antimicrobial Activity of Chilean Propolis

An antibacterial test was carried out in order to quantify MIC in bacteria cultures supplied with the same initial concentrations of PCP1, PCP2 and PCP3 (100 mg mL^−1^), as shown in Table 2. The MIC values were lower on the antimicrobial test for *S. aureus* in PCP2 and PCP3. PCP2 has 5 mg mL^−1^ with statistical differences in comparison with means of MIC (PCP1/ PCP3) that were 15 mgmL^−1^. Propolis samples against *E. coli* showed similar MIC values in PCP1/PCP2 and 30 mgmL^−1^,with statistical differences in PCP3 sample (as shown in Table 3).

## 3. Discussion

Propolis is a complex mixture of the beehive produced by the honeybee (*Apis mellifera*), who collects and transforms the bud exudates, by mixing them with waxy substances, used in the sepsis of the hive. This study aimed to investigate the botanical origin of propolis by conducting microscopic analysis of the pollen grains, and allowed us to identify different species of plants in three samples collected in the same season. The diversity of its biological activity has been related in numerous research studies to the activity of some components present in the extracts, especially certain types of polyphenols such as flavonoids.

Previously, a seasonal effect (time of collection) was demonstrated for the composition of chilean propolis, showing changes in polyphenol families identified in the samples where they were observed [16,17,18]. The influence of the area of collection determines the chemical composition of the honeybee products such as propolis, but in subtropical and tropical regions it is not possible to find *Populus* species in the vegetable sources and the propolis samples contain small amounts of flavonoids and major molecules such as polyprenylated benzophenones [12,19].Nevertheless, some investigations found a relationship between the common botanical origin of propolis and their similar chemical profiles; for example, in an apiary network of Central Chile, predominant species were *Salix humboldtiana*
*sp.* and *Eucalyptus*
*globulus,* among endemic and introduced plants, respectively [20,21]. Montenegro et al. found that the predominant species in the pollen grains in the Maule Region was *Cryptocarya alba*. They reported as the predominant botanical resources in the Metropolitan Region the *Quillaja saponaria*,*Salix *sp.*, Baccharis *sp.*, Buddleja globosa* and *Peumus boldus*; and native species such as *Aristotelia chilensis*, *Eupatorium glechonophyllum* and *Colliguaja odorifera*. In the Araucania Region, some studies described the species *Lotus ulginosus* as the main floral resource [17,22,23].

The propolis produced in Brazil, according to their botanical origin, come from *Hyptis divaricata*, *Baccharis dracunculifolia* and *Populus nigra* (Baccharis type propolis). Various authors have concluded that bees in non-temperate zones find other plant sources to replace poplar species. *Populus spp*. and their hybrids are the main sources of the propolis obtained in temperate zones (Europe, North America and the non-tropical regions of Asia) as reported in some studies [1,12].

The poplar-type propolis has a specific chemical composition where flavonoids and phenolic acid esters are the predominating groups of compounds. The main flavonoids described were flavones, flavonols (quercetin derivates), flavanones (pinocembrin derivates) and dihydroxyflavones such as daidzein [3,11].

In PCP1 and PCP3 we found that *Eucalyptus *sp.** pollen grains were abundant (Table 1). This species was no reported in previous studies; however the botanical origisn of Brazilian poplar propolis include *Poppulus*
*sp.* (Salicaceae); *Pinus*
*sp.*; alder (*Alnus glutinosa*), horse chestnut (Aesculus *hippocastanum*); elm (*Ulmus sp*.); oak (*Quercus sp*.) and beech (*Fagus *sp.*).* The species *Baccharis trimera* was identified in PCP2 and PCP3, and *Salix babylonica or Salix *sp.** pollen grains were identified in the three samples of chilean propolis, and this botanical origin determines the presence of flavonoids identified as quercetin and pinocembrin, observed in propolis obtained in temperate zones [24].

Honey bee propolis collected in Brazil was obtained from 11 species of the Apidae family: *A. andreniformis, A. binghimi, A. breviligua, A. cerana, A. dorsata, A. florea, A. koshevnikovi, A. laboriosa, A. mellifera, A. nigrocinta and A. nulesis* [14].

On the other hand, some authors argue that the influence of season in propolis collection cannot modify the quantity of phenolic compounds, but individual phenolic compounds can change. Finally, those differences determine the color of the propolis samples, such as: yellow, green, and red to dark brown [12].

In addition, the antioxidant activity of propolis was related to its geographic origins. Season is also an additional factor that may modify the chemical composition of propolis from the same area of collection, with similar vegetable sources. Additionally, in propolis from the same region, the botanical origin of the plants was determinant of chemical composition [11,16,19].

Moreover, biological properties, such as antioxidant activity, which is related to propolis chemical composition, can change with the season of collection; the total content of polyphenols and flavonoids in propolis was different. In south America, different compounds have been identified from propolis (HBP, honey bee propolis) [14,18].

Colorimetric assays to quantify total polyphenols in chilean propolis samples established that PCP2 (Angol sample) showed high quantities of phenolic compounds with significant statistical differences in comparison with the other samples (Figure 1). Flavones and flavonols quantified by AlCl_3_ methodology represented between 36.0% to 43.6% of total phenolic compounds.In contrast, the percentage of flavanones and dihydroflavonols determined by 2,4-dinitrophenylhydrazine reaction was between 60.9% to 79.1% of total phenolic compounds.Some chemical analyses performed in propolis samples described different proportions when quantifying total flavonoids using aluminum chloride (AlCl_3_),forming complexes that absorb at 425 nm.In our study, flavanones and dihydroflavonols were more abundant than flavones and flavonols; a plausible explanation of that difference is that the maximum absorption of flavonoids strongly depends on the presence or absence of a double bond in positions 2–3 on the flavonoid skeleton.Another explanation to the case of why flavanones and dihydroflavonols were not quantified by AlCl_3_ reaction is that, obviously, compounds do not contribute to the spectrophotometric measurements at 425 nm. Chromatographic and spectrometric studies aimed at quantifying total compounds showed differences in the results in previous studies, where calibration curves were performed with caffeic acid and not gallic acid or catechin and not using a mixture of pinocembrin–galangin (2:1 proportion) using the Folin–Ciocalteu method. These flavanones and dihydroflavonols are the main components in poplar-type propolis, as described by several authors [3,15]

Tannins and anthocyanins were quantified in chilean propolis, but their content was low in comparison with total polyphenols (Figure 2). Tannins (proanthocyanidins) were studied in Brazilian propolis to determine their content and correlation with total phenolic compounds. The highest tannin content, found in red propolis, was produced from leguminous tree species such as *Stryphnodendron adstringens* and *Acacia mearnsii*. Other plants from the Asteraceae family (*Baccharis dracunculifolia*), source of green propolis, contain substantial amounts of proanthocyanidins. Tannins and anthocyanins did not show statistical differences among chilean propolis samples; some authors found these types of compounds in propolis, as condensed anthocyanidins or leucoanthocyanidins [7]. Anthocyanidins were described as being found in natural sources, but not in propolis. In addition, phenolic acids and anthocyanidins were related to antioxidant properties [25,26].

The proportion of tannins in the chilean extract was between 15.5 to 20.2%, in contrast with 1 to 4% found in Brazilian propolis [23]. Due to the lack of information regarding the content of these compounds in chilean propolis, it is necessary to carry out chromatographic studies to confirm their content.

Recent chromatographic studies using LC–DAD–MS analysis have confirmed that propolis from different geographic areas showed differences in the types of compounds identified.Propolis collected in European countries, Argentina and China were characterized by the presence of phenolic acids and flavonoids [1,3,7,16]. The most abundant individual compounds were chrysin, pinocembrin, pinobanksin and galangin. Parallel studies in Brazilian propolis showed phenylated derivatives of p-coumaric acids, artepillin C, different caffeoylquinic acids and lower amounts of flavonoids, as the main groups. Individual phenolics identified were different from poplar-type propolis; they included, for instance: p-coumaric acids, phenylated cinnamic acids, ferulic acids and CAPE [12,25].

In the chilean propolis samples PCP1, PCP2 and PCP3, we identified and quantified among the main compounds: pinocembrin, quercetin and CAPE.This composition is similar to the composition of poplar-type propolis from temperate zones. (The results are shown in Figure 3 and Table 2). Poplar propolis was classified by Bankova and its chemical composition is different: Poplar propolis from Europe, North America and non-tropical regions of Asia contain flavones and flavanones and on the other hand green propolis from Brazil contains prenylated phenylpropanoids and caffeoylquinic acids from plants of the species *Clusia*. Flavanone, flavone and phenylpropanoid esters with high levels of pinocembrin and CAPE were obtained from *Populus *spp.** in propolis collected in Africa, Asia, America and Oceania [14,27].

Some investigation have been conducted to study individual flavonoids, in order evaluate their biological characteristics.Caffeic acid phenethyl ester (CAPE) has been found to play an important role in the apoptosis and cell cycles [7].

The antimicrobial potential of chilean propolis has been studied using different bacteria species. In our study, the MIC values observed in several assays of chilean propolis of PCP2 against *S. aureus* and *E. coli* were lower than the values of PCP1 and PCP3, with statistical differences (Table 3).

Chilean propolis exhibited antibiofilm potential and antimicrobial activities when it was evaluated on *Streptococcus mutans* cultures [28,29]. The antimicrobial activities of propolis with aromatic acids (ferulic, cinnamic, caffeic and p-coumaric acids) was described as having strong antibacterial actions against Gram-positive and Gram-negative bacteria [30].In addition, polyphenols from propolis showed anti-inflammatory activity, could act in the synthesis of matrix metalloproteinases and were identified as a potential target for osteoarthritis treatment and for reducing nociceptive pain in animal models [31,32,33].

Quercetin-3-glucuronide, a derivative of quercetin, and quercetin and isorhamnetin, can affect the transduction process of nuclear factor NF-kB, and consequently can decrease Nrf2 gene expression and allow the inactivation of miR-155 with pro-inflammatory activity [34]. Individual polyphenols such as CAPE, pinocembrin and apigenin can act on breast cancer cells, with antioxidant and anti-angiogenic properties [35,36,37].

The differences reported regarding antioxidant activity and differences in antimicrobial potential could be explained by the changes in the content of total polyphenolic groups and individual compounds in relation to the botanical origin of the samples. However, it is necessary to perform deeper studies to determine these factors.

## 4. Materials and Methods

### 4.1. Preparation of Extract of Chilean Propolis (PCP1; PCP2; PCP3)

In order to evaluate the effect of polyphenols from propolis against microbial activity and biofilm formation, propolis was collected during the spring of 2018 in three different climatic zones of Chile. The crude sample of propolis was kept frozen (−20 °C) and later crushed in cold; 100 g were dissolved in 100 mL of ethanol (70%) and macerated for 7 days at room temperature. The hydroalcoholic extracts of propolis were filtered with Whatman filter paper 2 and centrifuged at 327 g for 20 min at 5 °C.Finally, the solvent was evaporated at a temperature of 40 °C for 2 h in a rotary evaporator (Rotavapor Buchi R-210, Germany) and dissolved for 24 h with sterile DMSO (0.01%), in order to obtain 50% *w*/*w* concentrated extracts of chilean propolis (PCP1 from Angol region; PCP2 from Maule region and PCP3 from Melipilla).

### 4.2. Botanical Origin of Propolis

Propolis extract samples of 200 mg were homogenized and diluted in a mixture of: 1 mL of ethanol + 1 mL of chloroform + 1 mL of acetone.Later, three samples were centrifuged at 500 g and the solid was decanted. The sediment obtained in each sample was diluted in potassium hydroxide (10%), and then was treated at 100 °C for 2 min. The centrifugation and decantation procedure was repeated. The sediment obtained was added to 10 mL of absolute ethanol (95%). The centrifugation and decantation procedure was repeated. The sediment was observed at 400X magnification in an optical microscope. Subsequently, plant structures (pollen grains) were counted and identified. Identification was performed by comparing the different structures with photographs and permanent preparations available in the Ethology Laboratory of the Universidad de la República, Uruguay, and complemented with relevant literature. The proportion of each of the total structures counted weas estimated using melissopalynology analysis [38].

### 4.3. Determination of Total Phenolic Content in Extracts of Chilean Propolis

The content of total polyphenols in PE was quantified by Folin–Ciocalteu method. For this assay, 100 μL of hydroalcoholic extract of propolis, 100 μL of distilled water and 2 mL of Folin–Ciocalteu reagent (Merck, Germany) were mixed together. The solution obtained was incubated for 8 min and, finally, 3 mL of sodium carbonate 20% (*w*/*v*) were added. The absorbance of that solution was measured at 760 nm in UV/VIS 7305 Jenway spectrophotometer after 2 h of incubation at room temperature. The concentration of polyphenols was calculated from a calibration curve and was expressed in mgg^−1^ equivalent to the pinocembrin–galangin standard mixture 1:1 [39].

### 4.4. Flavones and Flavonols (Total Flavonoids) Content

These flavonoids were determined using the aluminum trichloride (2%) reaction, diluting 2 mL of the extract in pure ethanol (1:10 *v*/*v*), then 5 mL of AlCl_3_ were transferred to a 25 mL flask. The absorbance was measured at 425 nm in UV/VIS7305 Jenway spectrophotometer. The total flavones and flavonols content was expressed in equivalents to mgg^−1^ of quercetin standard by interpolation of the calibration curve using 50 to 1000 ppm of the standard solutions (Sigma-Aldrich, Germany) [39].

### 4.5. Total Flavanones and Dihydroflavonols

These compounds were quantified by means of the reaction with 2,4-dinitrophenylhydrazine. For this purpose, 2 mL of PE were dissolved in ethanol (1:10 *v*/*v*) and 2 mL of 2,4-dinitrophenylhydrazine were added (1%: 1 g of dinitrophenylhydrazine in 2 mL of sulfuric acid at 96% and completed to 100 mL with methanol). The absorbance was measured at 486 nm in a UV/VIS7305 Jenway spectrophotometer. The total flavanones and dihydroflavonols were expressed in equivalents to mgg^−1^ of pinocembrin by interpolation of the calibration curve using 50 to 1000 ppm of the standard solutions (Sigma-Aldrich, Taufkirchen, Germany) [39,40].

### 4.6. Quantification of Total Anthocyanins in Chilean Propolis

Total anthocyanins were quantified using the differential pH method; for that assay 50 µL of PE were added with sodium acetate buffer (0.025 M, pH 4.5) or potassium chloride buffer (KCl, 0.025 M, pH 1.0) in 25 mL flasks and then allowed to stand for 15 min. The absorbance was determined at 510 and 700 nm using the UV/VIS7305 Jenway spectrophotometer. Total anthocyanins were expressed as milligrams of cyanidin-3-glucoside (c-3-gE) × 100^−1^ g of propolis according to the formula in reference [41].

### 4.7. Quantification of Total Tannins in PCP1, PCP2 and PCP3 Samples

The quantification of total tannins is based on the property of tannins to precipitate proteins. First, we determined total polyphenols by the Folin–Ciocalteu procedure. After that, tannins were removed by precipitation with a solution of bovine serum albumin (BSA) (buffer 0.2 M acetate, pH 5.0; sodium chloride 0.17 M and 1.0 mg mL^−1^). Later, 1 mL of the extract and 1 mL of BSA solution were added. After that, the mixture was centrifuged at 5000 rpm for 15 min at room temperature. Finally, we again determined the total polyphenols using an aliquot of the supernatant (100 μL) by the Folin–Ciocalteu procedure. The content of total tannins was expressed as mgg^−1^ of gallic acid (GAE, mg gallic acid g dry material). The same curve was used to determine total polyphenols [42].

### 4.8. Balsam Content in PCP1, PCP2 and PCP3 Samples

From each crude sample, three parallel extracts using ethanol 70% (*w*/*w*), 2 mL propolis extract were evaporated to dryness until they reached a constant weight, and the percentages of balsam in the extracts were calculated as the ethanol soluble fraction. The mean of the three values was determined [1,39,40].

### 4.9. Identification and Quantification of Some Compounds Present in Chilean Propolis

Four flavonoids were identified and their concentrations were calculated by the direct injection method through a high-resolution liquid chromatograph (Shimadzu, Japan), equipped with an LC-20AT pump connected to a UV-Visible detector SPD-M20A UV (HPLC-DAD). The separation was carried out in a LiChrospher RP-18 column, with particle size 5 µm × 250 mm and stove CTO-20AC at 25 °C. The elution was carried out at 40 °C using a mixture of acetonitrile, methanol, water and 5% formic acid in a flow of 1000 mL min^−1^, with a gradient from 30% to 70% and 20 µL of the diluted PE sample (1:50) were injected. To calculate compound concentrations, we used solutions of the commercial standards at 5 ppm of apigenin, quercetin, pinocembrin and caffeic acid phenethyl ester (CAPE) (Sigma-Aldrich, St Louis, MO, USA), and the concentrations were expressed in mg L^−1^ by the interpolation of the calibration curve [39].

### 4.10. Antimicrobial Activity of Chilean Propolis (PCP1, PCP2, PCP3)

We used the serial dilution method, following the NCCLS guidelines [15]. The strains of *Staphylococcus aureus* ATCC 29213 and *Escherichia coli* ATCC 35218 determined the Minimum Inhibitory Concentration (MIC). The bacterial suspension 5 × 10⁵ CFU mL^−1^ was inoculated in 96–well microplates containing 100 μL of RPMI 1640 medium (Life Technologies TM), supplied with 100 μg mL^−1^ of PCP1, PCP2, and PCP3 in DMSO (0.1%). We used a control without propolis as a negative control (vehicle). All the tests were run in triplicate [43].

### 4.11. Statistical Analysis

Statistical analyses were performed using computational software. The Shapiro–Wilk test was applied to determine the normal distribution of the results. Experimental values of polyphenol concentrations and MIC means obtained from antimicrobial test were estimated by statistical analysis of variance (ANOVA) and Tukey post-test. Differences were considered statistically significant at *p*-values of 0.05 (*p* < 0.05).

## 5. Conclusions

These results suggest that the geographic area of collection determines the botanical origin and chemical composition of chilean propolis. The total polyphenols content and the groups of individual compounds and their concentrations can modify the antibacterial properties of the extract. However, it is necessary to carry out more chromatographic studies to confirm the exact content of phenolic compounds quantified by means of colorimetric assays.Studies that determine the influence of the geographic and floral variables in the chemical composition of propolis are a valuable source of information for the study of its biological properties.

## Figures and Tables

**Figure 1 plants-10-01543-f001:**
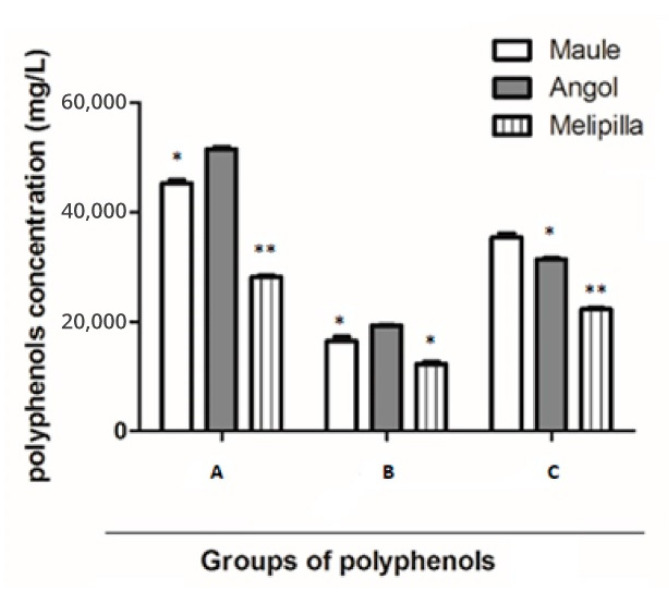
Total compounds quantified by means of colorimetric analysis in PCP1, PCP2 and PCP3. (**A**). Total polyphenols. (**B**). Total flavones and flavonols. (**C**). Total flavanones and dihydroflavonols. Statistical differences: (* *p* < 0.05), (** *p* < 0.01).

**Figure 2 plants-10-01543-f002:**
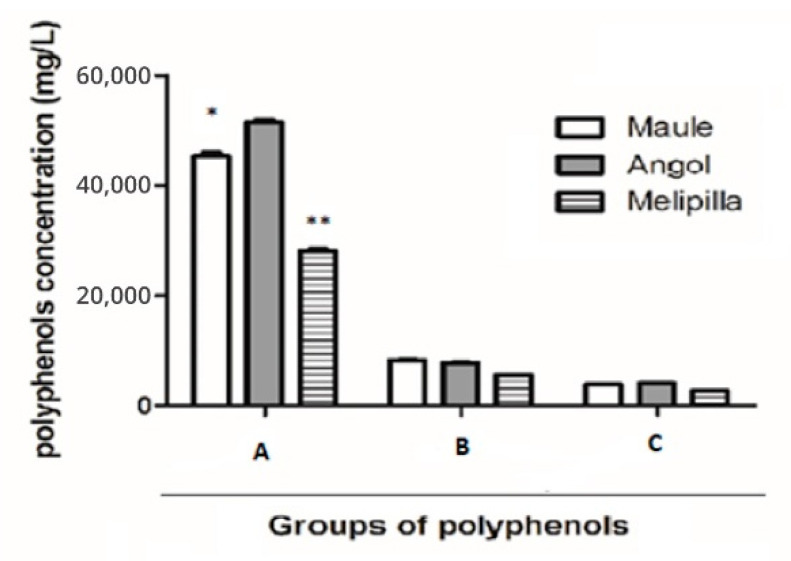
Total compounds quantified by means of colorimetric analysis in PCP1, PCP2 and PCP3. (**A**). Total polyphenols. (**B**). Total tannins. (**C**). Total anthocyanins. Statistical differences: (* *p* < 0.05), (** *p* < 0.01).

**Figure 3 plants-10-01543-f003:**
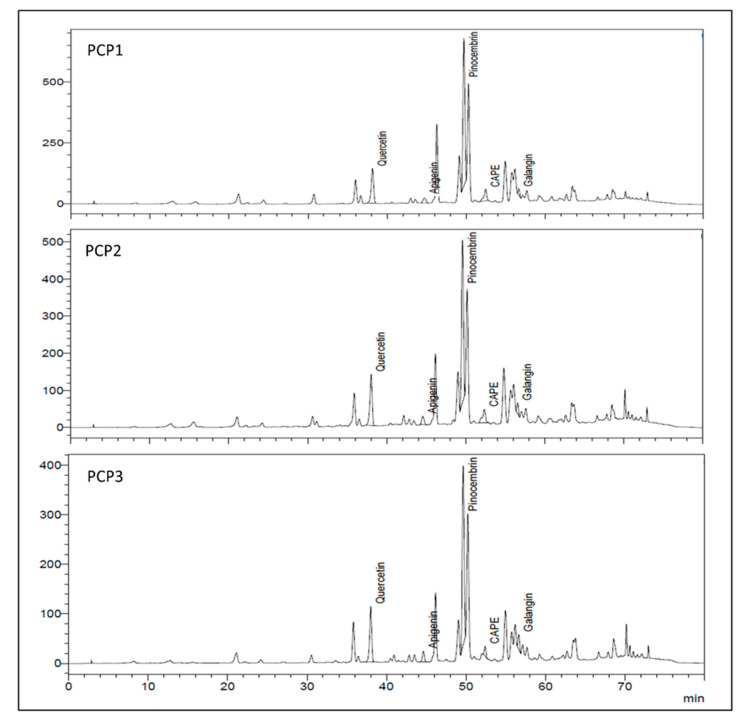
HPLC chromatograms of PCP1, PCP2 and PCP3. Concentrations of phenolic compounds were expressed in mg L^−1^.

**Table 1 plants-10-01543-t001:** Floral composition of chilean propolis PCP1, PCP2 and PCP3 samples.

PCP1		PCP2		PCP3	
Predominant Species	%	Predominant Species	%	Predominant Species	%
*Lotus *spp.**	27.9	*T. Scutia buxifolia*	12.3	*Raphanus raphanistrum*	19.5
*Fabaceae*	19.1	*Quillaja saponaria*	8.2	*Eucalyptus *spp.**	13.8
*Eucalyptus *spp.**	11.8	*Melilotus *sp.**	7.6	*Lotus *spp.**	12.6
*T. Mentha x piperita*	9.6	*Senecio *sp.**	7.2	*T. Trifolium repens*	5.7
*Melilotus *sp.**	7.4	*Pinus *sp.**	7.2	*T. Scutia buxifolia*	5.3
*Apiaceae*	5.1	*Eucalyptus *spp.**	6.8	*Salix *spp.**	4.5
*Echium plantagineum*	4.4	*Poaceae*	6.8	*Melilotus *sp.**	3.3
*Poaceae*	4.4	*Genista monspessulana*	5.5	*Senecio *sp.**	3.3
*Raphanus raphanistrum*	2.9	*T. Trifolium repens*	4.7	*Ligustrum lucidum*	3.3
*T. Cichorium intybus*	2.4	*Lotus *spp.**	3.9	*Scrophulariaceae*	2.8
*Pinus radiata*	2.2	*Raphanus raphanistrum*	3.1	*Polygonaceae*	2.4
*Salix babylonica*	1.5	*Salix babylonica*	2.9	*Poaceae*	2.0
*Senecio *sp.**	0.7	*Rosaceae*	2.9	*T. Acacia bonariensis*	1.6
*Acacia *sp.**	0.7	*T. Acacia bonariensis*	2.6	*Apiaceae*	1.6
		*Sorghum *spp.**	2.5	*Zea mays*	0.8
		*Atriplex imbricata*	2.1	*T. Baccharis trimera*	0.8
		*T. Peltophorum dubium*	1.9	*T. Peltophorum dubium*	0.4
		*T. Baccharis trimera*	1.3	*Salvia officinalis*	0.4
		*Apiaceae*	1.1	*T. Cynara cardunculus*	0.4
		*T. Cynara cardunculus*	0.8	*Casuarina cunninghamiana*	0.4
Not identified (*n* = 1)	0.1	Not identified (*n* = 2)	8.6	Not identified (*n* = 5)	15.1
Total	100	Total	100	Total	100

**Table 2 plants-10-01543-t002:** Individual polyphenolic compounds quantified by HPC-DAD in three samples of chilean propolis.

Compounds	Tr. (Minutes)	Concentrations ± S.D (mg L^−1^)		
		Propolis Samples		
		PCP1	PCP2	PCP3
Apigenin	44.5–44.6	8.5 ± 0.3	7.7 ± 0.1	6.2 ± 0.2
CAPE	52.3–52.5	16.8 ^a^ ± 0.5	14.2 ^a^ ± 0.4	4.2 ^a^± 0.7
Pinocembrin	49.4–49.7	184.7 ^b^ ± 0.5	134.2 ^b^ ± 0.3	107.3 ^b^ ± 0.3
Quercetin	30.5–38.0	127.9 ^b^ ± 0.4	120.4 ^b^ ± 0.2	88.8 ^b^ ± 0.1
Galangin	56.5–56.6	5.9 ± 0.6	6.8 ± 0.4	7.0 ± 0.3

Concentrations of compounds were expressed in mg L^−1^ as mean ± standard deviation. The *p*-value was calculated using ANOVA Multiple Comparison and the Tukey post-test. Statistical differences: (*p* < 0.05 ^a^; *p* < 0.01 ^b^).

**Table 3 plants-10-01543-t003:** Antimicrobial activity of polyphenols from Chilean propolis in PCP1, PCP2 and PCP3 on bacterial cultures.

Microorganisms		PCP1	PCP2	PCP3
*S. aureus*	ATCC 35218	15 ^a^ ± 0.3	5 ^a^ ± 0.3	15 ^a^ ± 0.3
*E. coli*	ATCC 29213	15 ^b^ ± 0.2	15 ^b^ ± 0.4	30 ^b^ ± 0.2

MIC: Minimum Inhibitory Concentration. MIC values were expressed in mg mL^−1^ as mean ± standard deviation. *p*-value was calculated as significant differences after ANOVA Multiple Comparison and Tukey post-test. Statistical differences: (*p* < 0.05 ^a^; *p* < 0.01 ^b^).

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
