# Peer review of "Geographic Area of Collection Determines the Chemical Composition and Antimicrobial Potential of Three Extracts of Chilean Propolis"

_plants, 2021, doi:10.3390/plants10081543_

Round 1

Reviewer 1 Report

  • In the title they mentioned about antibacterial potential! They weren’t presented in the abstract as results!
  • The authors had to introduce the interest of the polyphenols found in propolis in the introduction and give some examples! They had to highlight the biological and pharmaceutical interest! Same remark concerning the presentation the interest of propolis as an antibacterial product!
  • The authors have to pay attention to the coherence of the text; the first table is entitled table 4; there two table (number 2); they have to identify the % used in the legend (what exactly is it?). This legend in question (table 4) should be much better descriptive.
  • The authors have to watch out for the abbreviation! The first appearance in the text the name should be complete and in brackets the abbreviation which will then be presented in the rest of the manuscript.
  • It seems to me that the concentration which has antibacterial activity is a little high! This should be discussed thoroughly in the manuscript! Is it a significant activity?

Reviewer 2 Report

Comments to the Author:

the authors have clearly defined their line of conduct of the work and the result is interesting. But there are some comments and suggestions.

1) One of the three places mentioned (Melipilla) is near the metropolitan area of ​​Santiago de Chile. Has a pollution assessment been made in the area? Does this affect the quality of the pollen?

2) The purpose of the work is not clear in the introduction. A phrase must be added explaining the purpose of the work and the expected results.

3) Do the predominant floral species in the characterization of propolis correspond to the predominant species in the geographical area?

4) Use only S. aureus and e. coli for the evaluation of the antibacterial activity is reductive. Other lines could be considered.

5) Given that the seasonal effect was cited as very relevant, the reference season for the collection of the propolis under study must be mentioned.

6) The text has numerous typos.

7) in the caption of figure 3 the references PCP 2 and 3 are missing.

8) In figure 3, quercetin presents baseline in all three spectra. How is it possible? If quercetin was detected with a different methodology it must also be made clear in the caption and in the text.

Reviewer 3 Report

The paper "Geographic area of collection determines the chemical composition and antimicrobial potential of three extracts of Chilean propolis" described the biological activities ( antibacterial, antifungal and antibiofilm) of Chilean propolis from different regions and the differences in the chemical composition associated to the biological properties.

The work is well written and the esperimental methods are described in detail. In my opinion, the paper il suitable for the publication in "Plants"

Only a minor english editing is recommended.

Author Response

Thank you very much for yours comments an evaluation. The text will be improved in terms of it writing in English lenguage. Please see the attachment.

Round 2

Reviewer 2 Report

All responses were satisfactory.